# Survival analysis of pathway activity as a prognostic determinant in breast cancer

**Gustavo S. Jeuken**[1], **Nicholas P. Tobin**[2], **Lukas Käll**[1] *

**1** Science for Life Laboratory, School of Engineering Sciences in Chemistry, Biotechnology and Health, KTH – Royal Institute of Technology, Solna, Sweden, **2** Department of Oncology and Pathology, Karolinska Institutet and University Hospital, Stockholm, Sweden

\* lukas.kall@scilifelab.se

**Data Availability Statement:** All data used in this paper, except for the METABRIC dataset, can be found in https://github.com/statisticalbiotechnology/metabric-pathway-survival/tree/main/data The METABRIC data has been deposited at the European Genome-Phenome

## Abstract

High throughput biology enables the measurements of relative concentrations of thousands of biomolecules from e.g. tissue samples. The process leaves the investigator with the problem of how to best interpret the potentially large number of differences between samples. Many activities in a cell depend on ordered reactions involving multiple biomolecules, often referred to as pathways. It hence makes sense to study differences between samples in terms of altered pathway activity, using so-called pathway analysis. Traditional pathway analysis gives significance to differences in the pathway components' concentrations between sample groups, however, less frequently used methods for estimating individual samples' pathway activities have been suggested. Here we demonstrate that such a method can be used for pathway-based survival analysis. Specifically, we investigate the pathway activities' association with patients' survival time based on the transcription profiles of the METABRIC dataset. Our implementation shows that pathway activities are better prognostic markers for survival time in METABRIC than the individual transcripts. We also demonstrate that we can regress out the effect of individual pathways on other pathways, which allows us to estimate the other pathways' residual pathway activity on survival. Furthermore, we illustrate how one can visualize the often interdependent measures over hierarchical pathway databases using sunburst plots.

## Author summary

Most of the important cellular functions are carried out by not just individual biomolecules but are rather dependent on the concerted reactions involving large sets of biomolecules, which are referred to as pathways. Yet, measurement techniques naturally have to measure the abundances of each such molecule individually. To assess the difference in functional activity between samples one often uses statistical techniques to integrate abundances into pathway activity. Here we implemented a method for investigating which such pathway activities that are prognostic for patients' survival when analyzing breast cancers. We showed that the pathway activities are more prognostic of a patient's survival time than prognoses made directly from the measured concentrations of individual molecules. We also show which such pathway activities that are not just active due to the

Archive (http://www.ebi.ac.uk/ega/), which is hosted by the European Bioinformatics Institute, under accession number EGAS00000000083. The code for reproducing the analysis in this paper, including the code for generating its plots, is available at https://github.com/statisticalbiotechnology/metabric-pathway-survival.

**Funding:** This work has been supported by a grant from the Swedish Foundation for Strategic Research (BD15-0043) to LK. The funders had no role in study design, data collection and analysis, decision to publish, or preparation of the manuscript.

**Competing interests:** The authors have declared that no competing interests exist.

overall increased proliferation in malign cancers. We also illustrate how pathway activities can be efficiently and interactively visualized using so-called sunburst plots.

This is a *PLOS Computational Biology* Methods paper.

## Introduction

In molecular biology, high throughput experiments enable the measurement of thousands or even millions of analytes in any sample of biological origin. Such wealth of data makes it possible to describe samples very accurately, with a precision that opens the door to new understandings of the mechanisms governing biological and medical processes.

Although this abundance of data indicates a possibility for acquiring knowledge, the sheer number of dimensions in such measurements also presents us with many challenges: when analyzing high-dimensional vectors with statistical methods one easily faces the curse of dimensionality, i.e. that the sample space grows exponentially with each added dimension [1]. This becomes a problem when the number of samples is lower than the number of measured analytes. Further, in e.g. differential expression analysis, which tests the concentration differences of each measurement separately, we to some degree run into decreased sensitivity due to multiple hypothesis testing.

One of the more promising ways of alleviating these problems is through pathway analysis [2, 3]. Proteins frequently operate in an orchestrated manner and phenotypes are frequently the consequence of sets of proteins and not just of single-proteins. Metabolic pathways, or other aggregated biological knowledge that groups analytes, provide a model-driven way of combining molecular information and thus also the measurements from a high throughput experiment in a biologically meaningful way.

Traditional pathway analysis first determines the quantitative differences in gene expression between patient groups, and subsequently either tests the significantly differential genes annotation for enrichment in the tested pathway [4–6], or tests if the genes belonging to a pathway have more extreme differences than other genes, using so-called gene-set enrichment analysis [3]. Both types of analysis leave the user with a significance value for the analytes in a pathway annotation being differentially abundant under the different conditions. Yet this feels unsatisfying, as embedding the pathway analysis in the statistical test significantly limits the types of statistical tests that can be applied in the analysis.

An alternative is offered by single sample pathway analysis, which promises activity scores for each sample and pathway. The method ASSESS [7] fits two mixture models to data to quantify pathways, however, the supervised learning step of the procedure makes further statistical testing challenging. Another critique is that most pathway analysis methods derive statistical significance by comparing the behavior of genes belonging to a pathway with those that do not. Goeman & Bühlmann [8] argue in favor of self-contained tests, in which the significance of a pathway relates only to the expression of said pathway's genes, resulting in a more restrictive null hypothesis that leads to a higher statistical power. GSVA [9] uses a competitive test for gene set enrichment. The ssGSEA [10] method, by utilizing internal ranks of expression as a basis for enrichment, is thus also not self-contained. While this is also true for singscore [11], it has a large advantage in that it does not need other samples as a background when providing each sample's score.

Two methods, PLAGE [12] and Pathifier [13], will produce self-contained unsupervised metrics of pathway activity. The former does so through Singular Value Decomposition (SVD) in a space formed only by a pathway's genes. The latter takes the same approach but uses a Principal Curve instead, and while this enables non-linear gene interactions to be captured, it introduces the need for annotated baseline samples, as well as a larger dataset.

We believe in the merits of doing statistical analysis on a pathway level, as the function of a pathway is often much better understood than that of individual genes, and statistical operations on a pathway-level are often more directly related to the biology of the problem. Cell proliferation, as an example, has been established as one of the hallmarks of cancer [14, 15]. It is a complex, systemic process that involves many different mechanisms, in different parts of the cell, as well as signaling pathways that regulate them. In breast cancers, it has been shown that one can separate Luminal A from Luminal B subtypes based on the level of proliferation [16]. Thus, when investigating how molecular profiles of tumors affect patients, one may want to look at proliferation as a whole, instead of focusing on individual analytes.

However, the real advantage of assigning pathway activity to individual samples is that it opens the way for less blunt statistical analysis than case-control comparisons. Here, we reimplemented the PLAGE [12] for pathway summarization and studied its performance when applied to survival analysis using a Cox proportional hazards model [17]. We highlight the advantages and flexibility that working on a pathway level provides. Particularly we demonstrate how to counter the confounding effects on the pathway analysis of increased cell proliferation in severe breast cancers. We also demonstrate how one can use sunburst plots for exploratory visualization of the significance of pathway activities while maintaining a pathway hierarchy.

## Materials and methods

### METABRIC transcription profiles, and their annotation

Normalized gene expression and clinical annotation in the METABRIC dataset were downloaded from the European Genome-phenome Archive. This data consists of microarray reads from 1992 breast cancer specimens, primarily fresh frozen, together with the clinical annotation, including survival information, of the respective patients. Twelve samples were reported twice in the dataset [18] and these were removed from our analysis, but otherwise, the entire cohort was used.

Pathway annotations were retrieved from the Reactome database [19], version 76, with annotations as Ensembl gene IDs. The pathways' gene ID annotations were converted to Illumina probe IDs (HT_12_v4), through BioMart [20], by assuming that any transcript associated with each Reactome protein's underlying gene was associated with the pathway.

The clinical endpoint for this study was breast cancer-specific survival (BCSS) defined as patients who have not died from breast cancer in the study period from the date of surgery to the end of follow-up.

### Estimation of pathway activities

We followed the PLAGE [12] method's singular vector decomposition strategy. Let $\mathcal{G} = \{a_1, a_2, \ldots, a_g\}$ be the set of all measurements we want to study, the Illumina probes in our case, and let $\mathcal{P} = \{b_1, b_2, \ldots, b_p\}$, $\mathcal{P} \subset \mathcal{G}$ be the subset of measurements that are associated with our pathway of interest. We define $\mathbf{A}_{g \times m}$ as the matrix of log-transformed and standardized measurements of our samples (where $m$ is the number of samples), and $\mathbf{B}_{p \times m}$ as a matrix constructed by using only the rows in $\mathbf{X}$ that are present in $\mathcal{P}$. We then decompose $\mathbf{B}$ using a

Truncated Singular Value Decomposition (SVD):

$$\mathbf{B}_{p\times m} \approx \mathbf{U}_{p\times 1}\mathbf{S}_{1\times 1}\mathbf{V}^T_{m\times 1} \tag{1}$$

Note that there is a slight difference in preprocessing compared to Tomfohr *et al.* [12] which use a $\mathbf{A}_{g\times m}$ that contains standardized measurements that were not log-transformed. Here we name the pathway's left singular vector $\mathbf{U}$ as an eigensample, and the right singular vector $\mathbf{V}$ as an eigengene, following the nomenclature in Wall *et al.* [21]. Note that the eigengene contains one vector element per sample, and we will use these vector elements as a measure of pathway activity. Practically we here make use of the *scikit-learn* python package [22].

## Proportional hazards model

Cox's proportional hazards model [17, 23] relates the survival function $S_i(t)$ of patient $i$ to the value of any of its covariates $X_i$ as,

$$\lambda(t|X_i) = \lambda_0(t)\exp(X_i \cdot \beta). \tag{2}$$

Where $\lambda(t)$ is the hazard function and is defined as the rate of mortality at time $t$, for patients that have survived up to that time, and $\lambda_0(t)$ is the baseline hazard defined as $\lambda(t|0)$. The hazard function is then associated with the survival function by $\lambda(t) = -S'(t)/S(t)$.

Here we used the pathway activities derived previously as the explanatory variables $X$ for a Cox regression, doing so one pathway at a time, to study the connection between each of the samples' pathway activities and the survival of patients. This regression gave us both a coefficient $\beta$ showing the magnitude of the effect of a pathway, and a $p$ value representing the statistical significance of its coefficient. The $p$ values were subsequently corrected for multiple testing into $q$ values [24]. In here we make use of the *lifelines* python package [25].

## Concordance index

The concordance index (or C-index) is a generalization of the area under the curve (AUC) classifier performance that can take into account censored data. It represents the model's accuracy in ranking the survival times of the samples [26]. It can be calculated as,

$$c = \frac{\sum_{i\neq j}\mathbb{1}_{t_i<t_j}\mathbb{1}_{\eta_i>\eta_j}d_j}{\sum_{i\neq j}\mathbb{1}_{t_i<t_j}d_j}. \tag{3}$$

Here, for each patient $i$, we have the observed survival time, $t_i$, and the censoring variable, $d_i$, that takes a value of either 1 if the event of death has been observed and 0 otherwise. The indicator variable $\mathbb{1}_{t_i<t_j} = 1$ if $t_i < t_j$ and 0 otherwise. The variable $\eta$ is the hazard score for each sample, calculated as $\eta_i = X_i \cdot \beta$. The C-index is calculated using 5-fold cross-validation: for each step, the coefficients $\beta$ are fitted to 80% of the data, the hazard score $\eta$ is calculated for the samples in the holdout data and the index is obtained by comparing it to their survival status. Just as for AUC, a concordance index of 0.5 corresponds to a null prediction and 1 to a perfect prediction.

# Results

## A pathway-level survival analysis

We implemented a method for evaluating a pathway activity's influence on survival, based on a Cox's proportional hazards model [17] on top of the PLAGE method to estimate pathway activity from transcript abundances [12]. The method is generic for any censored data,

however, here we demonstrate its efficiency on the METABRIC breast cancer dataset. The transcription profiles of the 1980 breast cancers in the METABRIC data were downloaded and grouped according to the Reactome database into pathway groups. After the operation, each pathway group contained the transcripts corresponding to proteins in each pathway, and as there are overlaps between pathways in Reactome, this means that each transcript could appear in multiple pathways. The expression matrix of each pathway was factorized into eigensamples (left-singular vectors) and eigengenes (right-singular vectors) using the first eigenvector of a Singular Value Decomposition (SVD) following Wall *et al.* [21]. The eigensample represents the linear combination of genes and the eigengene the linear combination of samples that best explains the variance of the expression matrix of the pathway. This slightly backward naming convention stems from the idea that an eigengene models the variation of samples of a typical gene, while an eigensample models the variation of the genes in a typical sample.

We then project each sample into this eigengene and use this result as a representation of the sample's pathway activity. These pathway activities together with the survival information for each patient were fed into a Cox model for proportional hazards, which regresses these values against the survival information (survival time and disease-specific death) to obtain both the regression coefficient and the statistical significance for the effect of each pathway activity on patient survival. The significance, first obtained as $p$ values were multiple testing corrected into $q$ values [24]. Fig 1 gives an overview of the procedure.

Before we investigate the output of our model, we would like to motivate PLAGE's choice of SVD as a means to capture pathway activity. As an example, let's consider the pathway 'Metabolism of folate and pterines" consisting of 26 proteins. When investigating the covariation matrix of the transcripts and the survival time for 907 samples from patients that were

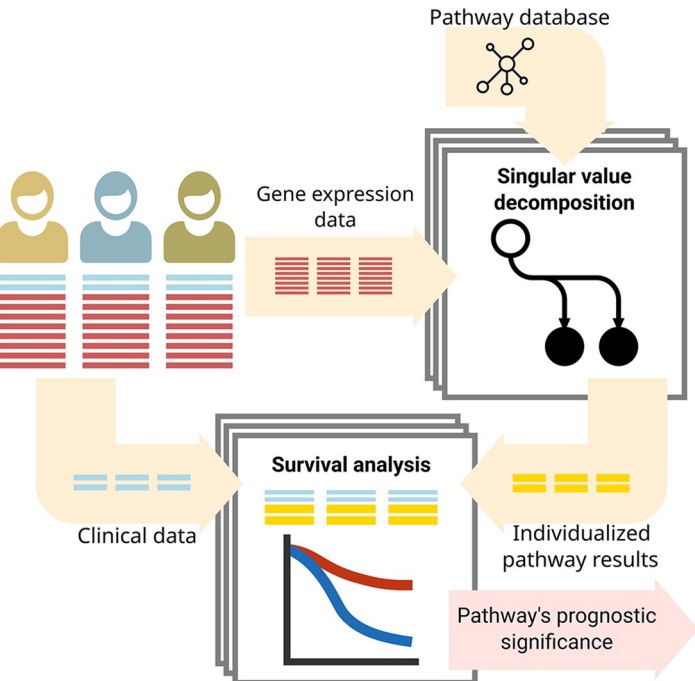

**Fig 1. A method to analyze the coupling of pathway activity on patient survival.** We used singular value decomposition to give, for each sample and pathway, an individualized pathway readout. We then combine those readouts with the survival information to perform survival analysis on a pathway level.

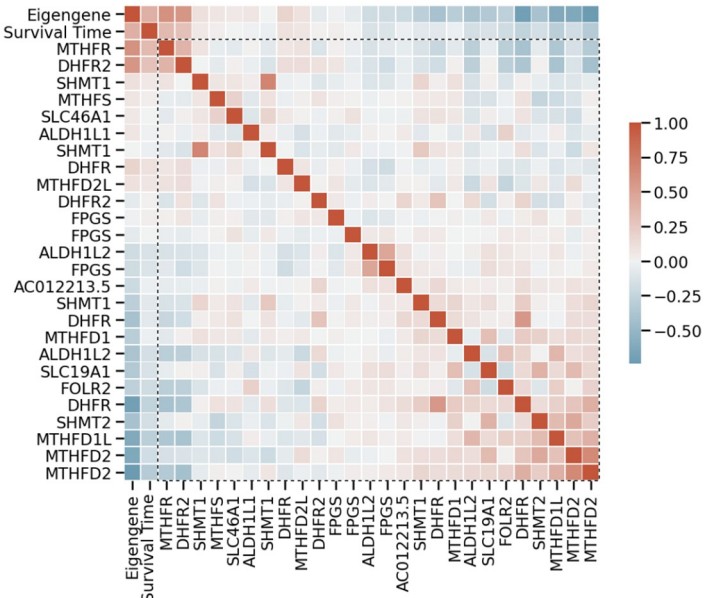

**Fig 2. Heatmap of the coefficients of the Pearson correlation matrix between the variables survival time, the eigengene of the gene expression values, and the gene expression values within the pathway 'Metabolism of folate and pterines'' for samples from diseased patients.** We sorted the genes in the expression matrix by their contribution to the eigensample and encompassed them with a dotted line. The patients' survival time both correlates and anti-correlates with the different genes. However, the eigengene captures the covariational trend within the expression data and correlates well with the patients' survival time.

deceased in the METABRIC cohort, we found both examples of transcript levels that correlate positively and negatively with the survival time (Fig 2). By having the probe reads ranked by their contribution to the eigensample, we see that the eigensample captures information from genes that have both a positive and negative influence on survival. We also see that probes that have a negative contribution to the eigensample also have a negative influence on survival, while the opposite is also true. It is the linear combination of these expressions, obtained in an unsupervised manner, that is used for later survival analysis.

## Analysis of breast cancer data set

The transcription profiles of the 1980 breast cancers in the METABRIC data were down-loaded, and they were analyzed with our method. S1 Table list the pathways and their prognostic significance, and the results are also available as an interactive plot (https://statisticalbiotechnology.github.io/metabric-pathway-survival/results.html). We found 1030 pathways (out of the 2214 pathways in Reactome) that were associated with patient survival with $q \leq 0.05$. We note that most pathways involved with the Cell Cycle are indeed associated with survival, as one would expect, as well as pathways relating to DNA replication and DNA repair.

This result is reassuring but possibly insipid, as we already know that Cell proliferation is a driver of cancer [14], and it specifically has a strong influence on survival. We also know that it affects most other aspects of cancer cell activity, making it hard to disentangle this signal from other important processes that affect the survival of patients. Luckily we have scored the activity of all pathways and can use them to highlight associations that otherwise are drowned by the more trivial background of cell proliferation.

Traditionally proliferation is quantified by the transcription of marker genes, like MKI67 [27]. However, here we instead used the pathway activity of the "Diseases of the mitotic cell cycle", calculated by our model, as a proxy for abnormal cell proliferation. We then regressed out the influence of proliferation on other pathway activities, by adding an independent variable for cell proliferation in our Cox's regression model for each pathway.

After regressing out our measure of proliferation the number of significant ($q \leq 0.05$) pathways, as expected, is reduced to 264 (S2 Table, (https://statisticalbiotechnology.github.io/metabric-pathway-survival/results_proliferation.html). We see that the operation removes the significance of most pathways relating to the cell cycle and DNA replication. Also, several other pathways' significance diminishes, e.g. the "Metabolism of folate and pterines" pathway became insignificant, indicating that its prognostic power was mostly driven by cell proliferation (when removing that confounding effect of cell proliferation, its effect on the survival of the patients is removed, from $q = 9 \cdot 10^{-13}$ to $q = 0.17$). However, pathways relating to cell pH regulation due to respiratory oxidation [28] (e.g. "Bicarbonate transporters", from $q = 2 \cdot 10^{-10}$ to $q = 7 \cdot 10^{-5}$) and HER2-signaling [29] (e.g. "GRB7 events in ERBB2 signaling", from $q = 3 \cdot 10^{-8}$ to $q = 7 \cdot 10^{-5}$) remain highly significant after the operation. This demonstrates an advantage of calculating individual measurements of pathway activity for each sample, as we can subtract the effects of known confounders.

## Pathway's eigensamples appear stable and their estimates appear well calibrated

For the results to be meaningful, the eigensample should be stable, i.e. if the decomposition is repeated for similar samples, a similar eigensample should result. This is especially important for datasets that are smaller than the ones used here. To simulate this, we randomly subsampled 20% (396 samples) of the tumors 100 times, each time performing the same decomposition, and made pairwise comparisons of the resulting eigensamples using the cosine distance (S1 Fig). We found that most eigensamples were indeed stable, and most importantly, the most significant pathways in respect to survival prognosis were all stable (S2 Fig).

We also performed a permutation test to check the calibration of the $p$ values coming from the Cox regression model. We randomly selected 100 pathways by their regression $p$ value, by first sampling $a$ uniformly in the range $[-5, 0]$ and selecting the pathway with the $p$ value closest to $10^{-a}$, then for each pathway the association between the gene expression values and the survival information was permuted $100/n$ times, where $n$ is the regression's $p$ value. The fraction of permutations with a more extreme outcome was then compared to the original Cox model's $p$ value. The results (S3 Fig) indicate that our model is indeed well-calibrated for the null hypothesis that there is no association between gene expression in the form of the pathway activation and patient survival.

Finally, to test whether the signal we see comes from a meaningful set of transcripts, as opposed to any set of transcripts, we performed a gene set permutation analysis (S1 Text), which confirmed the relevance of the manually curated sets.

## Pathways are more predictive of survival than individual transcripts

We also tested how well eigengenes can be used to predict the survival of each patient. We do this by, for each pathway, performing 5-fold cross-validation using the Cox model to predict survival on the holdout data, measuring the success of the prediction using the Concordance Index. Fig 3A shows the distribution of the Concordance Index after performing the cross-validation on all pathways and contrasts it to the distribution of the concordance index obtained by doing the same process with individual transcripts. We see that there is a general gain in

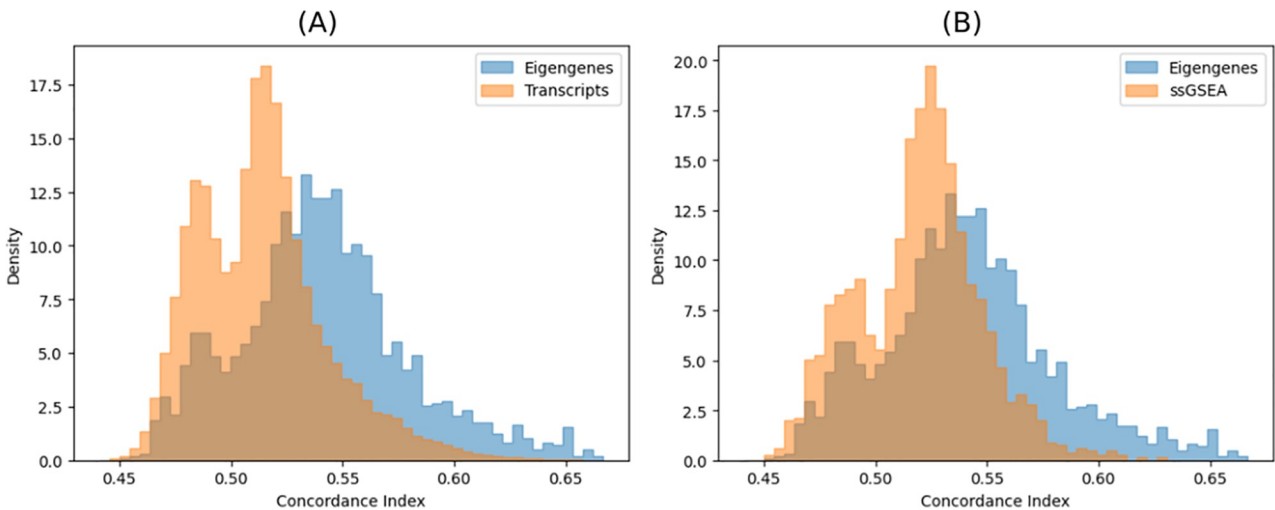

**Fig 3. Comparison of how well eigengenes predict survival.** A histogram of the concordance indices was calculated with 5-fold cross-validation of Cox regressions based on eigengenes compared to (A) individual transcripts and (B) ssGSEA enrichment scores.

predictive power when combining the information contained in individual transcripts using our method. When we compare pathways against only their constituent transcripts, we note that the presence of a strongly predictive transcript is a necessary but not sufficient condition for a highly predictive pathway, e.g. the top 10 best predictive pathways contain 6 of the top 10 transcripts, yet these same transcripts are present in pathways with no predictive power ($0.49 < CI < 0.51$). It is also interesting to note that in both cases there is a depletion of scores around the null prediction of 0.5, this deviation is further indication that transcripts, as well as pathway scores, are not independent.

## Eigengenes are more predictive of survival than enrichment scores

We compared our method to the alternative method of using the enrichment scores of single-sample geneset enrichment analysis (ssGSEA) [10] instead of eigengenes as an independent variable in a Cox-regression. To obtain the enrichment scores, we used ssGSEA as implemented in https://github.com/broadinstitute/ssGSEA2.0 with its default parameters. We found our eigengenes more sensitive than ssGSA, with 1030 vs 352 reported pathways with $q$ values below 5%. We also found that the concordance indices of our model outperformed the ones produced by the ssGSEA-fed model (Fig 3B).

## Including more than one eigengene makes the model more sensitive to covariates

We investigated an extended PLAGE model by including more than one eigengenes for each pathway. When checking the predictive performance of regressing all the eigengenes together, using a 5 fold cross-validation, there is a significant increase in the number of pathways that have high predictive power, when compared to a single eigengene (S4(A) Fig). This, however, is likely just a consequence of increasing the degrees of freedom of the regression, and we can see that when we regress each of the eigengenes separately, we do not see the same effects (S4(B) Fig). Similarly, when looking at the $p$ values for each coefficient obtained in the regression (S5(A) Fig), we find only a marginal gain when looking at more than one component at once.

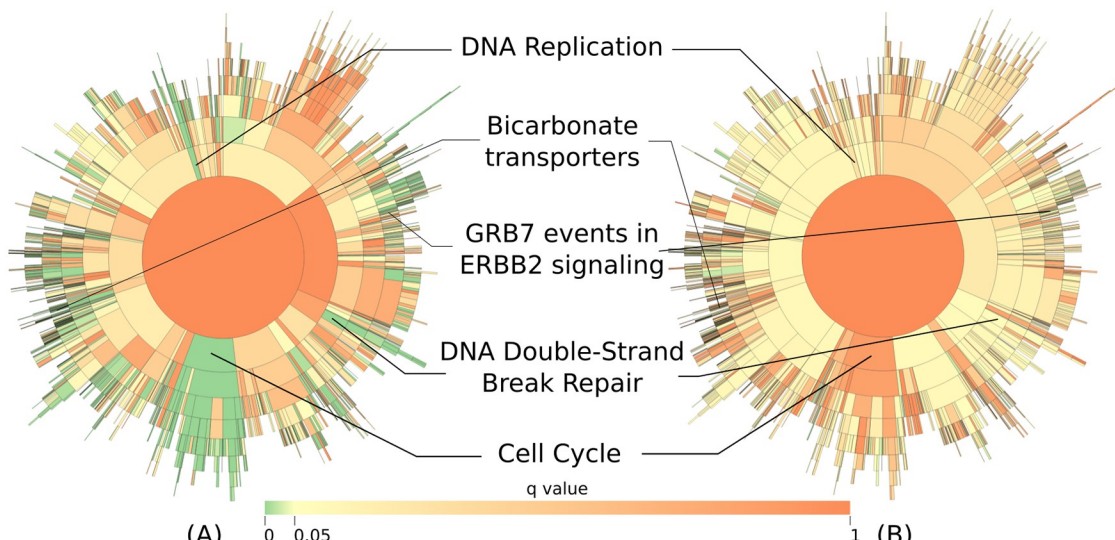

**Fig 4. Sunburst diagrams offer an exploratory visualization of the influence of the pathways in the Reactome hierarchy on the survival of patients.** We use an interactive sunburst chart to visualize and explore the result. This allows us to navigate the pathway hierarchy and see the relationships in a more natural way than e.g. in a static table. The sunburst (A) without and (B) with the pathway "Diseases of mitotic cell cycle" regressed out. Interactive versions of the same figures are available from https://statisticalbiotechnology.github.io/metabric-pathway-survival.

Looking at multiple eigengenes, however, also increases the exposure of the analysis to confounding factors. We noted that gene expression is not independent, and thus exogenous factors will influence the variance of the expression of genes inside the pathway. When adding more singular vectors to the analysis, the likelihood that one will capture such confounding factors increases. This becomes clear if we perform the regressions again, this time controlling for the effects of proliferation (S5(B) Fig), where we see that the advantages of using multiple eigengenes vanish and we start to see the burden of testing the extra hypothesis.

### Interactive visualization of pathway-level results

We know that biological pathways are not independent, yet the relationship between pathways also has a hierarchical nature, and in many pathway databases, large pathways are composed of smaller, connected pathways. This information is lost when we present a table with the results of our analysis, and for this reason, we developed an interactive visualization of the results based on a sunburst chart that highlights not only the numerical results but also the hierarchy and relationship between pathways. Fig 4 shows a static view of our interactive figure, the full plot is available at https://statisticalbiotechnology.github.io/metabric-pathway-survival/ Looking at pathway statistics in a hierarchical plot is a step in trying to understand the interactions between scores, and have a clearer picture of which biological processes are driving the significance in the tests.

### Discussion

Here we have reimplemented the PLAGE method for deriving a quantitative value of pathway activity for each sample, and we demonstrated its usefulness when associating pathway activity to patient survival. This is done by singular value decomposition of the expression of genes

inside a pathway, using the resulting eigensample to ascribe a pathway activity to each sample, and testing its association to the survival of patients using a Cox Proportional Hazards model.

Over the years databases of metabolomic pathways have been collected, condensing knowledge of molecular biology into causative relations. We here have made use of such relations, or at least their ability to group molecules in functional relevant subsets. The correlation of the concentration differences of such a subset of gene products can be seen as a manifestation of the same biological phenomenon. Thus, we argue that pathway-level statistics give a clearer view of how biological processes govern clinical outcomes.

When integrating the signals from multiple gene-products into one pathway activity, noise should be reduced as the random measurement errors tend to be averaged out, while the underlying common gene-regulation will corroborate over the analytes. Also, with pathway-level statistics we test fewer hypotheses than at the gene level, which increases the statistical power of the pathway-level measurements. Despite this, both over-representation analysis and gene set enrichment analysis are often found insensetive in terms of the number of significantly regulated pathways at a given false discovery rate.

The results of our analysis showed many pathways that are involved with cell proliferation as the most significantly associated with the survival of the patients. Although this is an obvious result, it is also a welcome find that validates our method. Furthermore, our strategy to extract individual measurements of pathway activity for each sample allows us to elaborate on the statistical model of survival. By regressing out the effects of proliferation, the confounding effect of this major perturbation of our samples was removed, allowing us to see other features of our data more clearly [30].

When adding multiple principle components, we obtain multiple measurements of pathway activity for each pathway and sample. In such expanded analysis one hopes that all the measurements capture the active regulation of the pathway, however, it also exposes the model to confounding factors. We hence find that PLAGE's usage of only the first eigenvector of its decomposition gives a well-found compromise between having a sensitive and stable measure and one that is a good representation of pathway activity.

One drawback of pathway analysis, however, is that, while gene expression is itself not independent, pathways by their nature overlap and so are intrinsically dependent entities, making true multiple hypotheses corrections an even less trivial task. One should hence make sure to also check the significances of the sub- and super-pathways of the examined pathway in the studied pathway database. We believe that the sunburst plots presented in this study are very powerful for such considerations.

At the heart of the method is the unsupervised dimensionality reduction that produces a pathway activation score for each sample. While we show some advantages of this idea on survival analysis, these should not be restricted to this field alone, and are suitable for application on any analysis where readouts are used to derive statistical significance.

Finally, SVD-based pathway analysis can be extended to include different modalities of data such as proteomics or metabolomics. As long as the novel data can be grouped into pathways, and the assumption of covariation within and between the different modalities hold, they can be included in the same factor analysis step and their information used for integrating pathway activity in an even broader sense.

## Supporting information

**S1 Text. A null model based on set permutation.**
(PDF)

**S1 Fig. Test of stability of pathways eigensamples.** We randomly selected a subset of 20% (398 samples) of the tumors and made pairwise comparisons of the direction of the eigensamples using the cosine distance. To provide a background measure, the procedure was repeated for a collection of random vectors directions picked uniformly in spaces with the same dimensions as the original data.
(TIF)

**S2 Fig. Pathways' significance as a function of their decompositions' stability.** We compare the results of the test of stability against the statistical significance derived from the regression of each pathway's activity against survival.
(TIF)

**S3 Fig. Investigation of the statistical calibration of $p$ values from the Cox regression model.** The associations between gene expression values and survival status were permuted and the fraction of permutations with a more extreme outcome was compared to the Cox model's $p$ value.
(TIF)

**S4 Fig. Comparing the predictive power of regressing more than one singular vector.** A) The concordance index distribution when building on joint regression model including a specified number of eigenes of the same pathway. B) The concordance index distribution when builing separate regression models for each eigengene.
(TIF)

**S5 Fig. Comparing the statistical power of testing more than one singular vector.** A) The fraction of tests that are under a certain $q$ value threshold, by regressing the $n \in [1, 5]$ fist singular vectors of each pathway together, as well as for testing individual transcripts. B) Same context, but now we control for the proliferation signal in each regression model.
(TIF)

**S1 Table. Significance of Cox-regression of Reactome's pathways.**
(TSV)

**S2 Table. Significance of Cox-regression of Reactome's pathways, when regressing out Diseases of the Mitotic Cell Cycle pathway.**
(TSV)

## Acknowledgments

This study makes use of data generated by the Molecular Taxonomy of Breast Cancer International Consortium, which was funded by Cancer Research UK and the British Columbia Cancer Agency Branch. The computations were enabled by resources provided by the Swedish National Infrastructure for Computing (SNIC) at HPC2N.

## Author Contributions

**Conceptualization:** Gustavo S. Jeuken, Nicholas P. Tobin, Lukas Käll.

**Data curation:** Gustavo S. Jeuken, Nicholas P. Tobin.

**Funding acquisition:** Lukas Käll.

**Investigation:** Gustavo S. Jeuken, Lukas Käll.

**Methodology:** Gustavo S. Jeuken, Nicholas P. Tobin, Lukas Käll.

**Project administration:** Lukas Käll.

**Software:** Gustavo S. Jeuken, Lukas Käll.

**Supervision:** Lukas Käll.

**Validation:** Gustavo S. Jeuken, Nicholas P. Tobin.

**Visualization:** Gustavo S. Jeuken.

**Writing – original draft:** Gustavo S. Jeuken, Nicholas P. Tobin, Lukas Käll.

**Writing – review & editing:** Gustavo S. Jeuken, Lukas Käll.

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
