## [Decision Letter · Decision Letter 0]

17 Aug 2021

Dear Dr. Käll,

Thank you very much for submitting your manuscript "Survival analysis of pathway activity as a prognostic determinant in breast cancer" for consideration at PLOS Computational Biology.

As with all papers reviewed by the journal, your manuscript was reviewed by members of the editorial board and by several independent reviewers. In light of the reviews (below this email), we would like to invite the resubmission of a significantly-revised version that takes into account the reviewers' comments.

We cannot make any decision about publication until we have seen the revised manuscript and your response to the reviewers' comments. Your revised manuscript is also likely to be sent to reviewers for further evaluation.

Sincerely,

Christos A. Ouzounis

Associate Editor

PLOS Computational Biology

Douglas Lauffenburger

Deputy Editor

PLOS Computational Biology

Reviewer's Responses to Questions

**Comments to the Authors:**

Reviewer #1: This paper presents an approach to generating biological pathway scores for individual samples and using them for survival analysis. This results in discovery of pathways which are significant predictors of survival. New approaches to pathway analysis are always welcome, especially those which generate single sample scores.

The paper is nicely written and mostly easy to follow. The work seems soundly performed. My main concern is over novelty and significance of the work as detailed below.

1. The scoring method seems very similar to another single sample pathway method PLAGE (Tomfohr, J., J. Lu, and T. B. Kepler. 2005., BMC Bioinformatics, 6: 225.). This method also uses SVD to obtain sample level pathway scores. From this it would seem the novelty of the current manuscript derives from coupling the SVD scoring method with conventional survival analysis.

2. It is a little unclear what the exact gap or need is being addressed by this method. In the introduction it is stated that the technique summarises pathways ‘distinctively from the statistical analysis”, by which I think they mean the outcome variable is not used to form the pathway scores (ie it is unsupervised). Is this the main objective? There are several other unsupervised single sample pathway methods, e.g. ssGSEA, PLAGE, singscore, MOGSA. I would encourage the authors to explain more clearly the motivation for their work.

3. Line 87 “pathway activities derived previously as independent variables X”. Clearly pathways overlap significantly and therefore the pathway scores will not be independent. How is this accounted for?

4. Why do the authors only used the first singular vector to summarise the pathway activity? What proportion of the variance is accounted for? Presumably large pathways may require more singular vectors to account for a similar proportion of variance and therefore the method more accurately summarises small pathways.

5. Equation 3: Please explain in a little more detail, as many readers may not be familiar with this. E.g. what is meant by survival status d? (Is this not the censoring indicator variable?)

6. Stability comparison: Main text says cosine distance was used to compare direction of subsample eigensamples with original eigensamples. But SI Fig 1 x axis label is “concordance index”. Additionally it is stated that “the most significant pathways in respect to survival prognosis were all stable”. Where is this shown? What effect did the subsampling have on survival prediction (e.g. distribution of concordance index)? The null represented by sampling expression values from U(0,1) seems a bit unrealistic.

7. Figure 3: The distributions look bimodal. Do the authors have any insight into why this is? Also do the best predicting pathways encompass the best predicting genes?

8. SI Note 1: The null simulation samples genes independently. Thus the resulting artificial pathways will be much more independent than the real pathways, which is rather unrealistic. In your real pathway set, you might expect more significant pathways since many of them are highly correlated. Instead, you could randomly shuffle the gene-pathway annotation matrix to obtain null pathways which maintain the level of pathway overlap in the real collection.

9. I could not find the code used to do the analysis.

Minor comments

10. Several typos or formatting problems, e.g. line 69 “PsubsetG”, lines 145 & 157 “q ! 0.05”. Also SI figure numbers are missing in main text.

11. Figure S2 caption: “associations between gene expression values were permuted”. I think you permuted the correspondence between gene expression values and the survival outcome? Or was each gene independently permuted?

Reviewer #2: In this manuscript Jeuken et al explore the prognostic value of pathway activities based on the transcription profiles of breast cancer of the METABRIC dataset.

I appreciated paper because well written and for its originality.

I have just some comments to do:

- Authors could better explain if used METABRIC breast cancer dataset overall, or selected patients according specific criteria;

- In addition, Authors could assess the prognostic value of pathway activities by grouping transcription profiles according breast cancer subtypes, in order to highlight the influence of the pathways on four different subtypes and their prognosis.

**Have the authors made all data and (if applicable) computational code underlying the findings in their manuscript fully available?**

Reviewer #1: **No: **Analysis uses public data and packages. However authors do not seem to provide scripts to reproduce the analyses.

Reviewer #2: Yes

PLOS authors have the option to publish the peer review history of their article (what does this mean?). If published, this will include your full peer review and any attached files.

Reviewer #1: No

Reviewer #2: No
---

## [Decision Letter · Decision Letter 1]

8 Dec 2021

Dear Dr. Käll,

Thank you very much for submitting your manuscript "Survival analysis of pathway activity as a prognostic determinant in breast cancer" for consideration at PLOS Computational Biology.

As with all papers reviewed by the journal, your manuscript was reviewed by members of the editorial board and by several independent reviewers. In light of the reviews (below this email), we would like to invite the resubmission of a significantly-revised version that takes into account the reviewers' comments.

We cannot make any decision about publication until we have seen the revised manuscript and your response to the reviewers' comments. Your revised manuscript is also likely to be sent to reviewers for further evaluation.

Sincerely,

Christos A. Ouzounis

Associate Editor

PLOS Computational Biology

Douglas Lauffenburger

Deputy Editor

PLOS Computational Biology

Reviewer's Responses to Questions

**Comments to the Authors:**

Reviewer #1: The authors have made a good attempt to address my comments. The acknowledgement of the similarity to an earlier method is welcome. However this does make the claim to novelty / significance somewhat weaker and harder to assess.

1. Thanks for acknowledging the similarity to PLAGE. The focus of the paper is now clearer, but the result is that the novelty seems less significant.

2. OK thanks for adjusting the introduction.

3. The authors admit that their method does not take into account correlations among genes and overlaps between pathways. They argue that other methods have similar shortcomings, which is true. The added discussion point is ok. But I would have liked to have seen some attempt at addressing this problem, rather than just pointing out it is a common issue, especially given the lower degree of claimed novelty in the revised manuscript. Finally, I think you mistyped ‘explanatory’ as ‘exploratory’ in the revised text.

4. I appreciate that more singular vectors results in a higher multiple testing penalty. But it seems to me more important to ask whether you are missing information by only using one singular vector. I think this could have been investigated, or at least commented on.

5. The extra text is helpful, thanks.

6. Thanks for fixing fig S1 and adding fig S2. For fig S2, I think you could use the q-value as the y axis so as the reader can see where the significance level was drawn. You mention modifying the text to clarify this point, but I could not see where this was. (The whereabouts of changes to the manuscript were not given in the response).

7. I think it could be useful to add a short comment on the bimodality to the text. The additional text regarding best-predicting pathways/genes is welcome.

8. The authors seem to say that this point does not matter because a) GSEA also has this problem and b) it is computationally easier. I don’t find this a very strong argument.

9. Although the github site is given in the text, nowhere does it say that the code is available there. I think this should be explicitly stated.

10. OK

11. OK

Reviewer #3: see attachment

**Have the authors made all data and (if applicable) computational code underlying the findings in their manuscript fully available?**

Reviewer #1: Yes

Reviewer #3: Yes

PLOS authors have the option to publish the peer review history of their article (what does this mean?). If published, this will include your full peer review and any attached files.

Reviewer #1: No

Reviewer #3: No
---

## [Decision Letter · Decision Letter 2]

15 Mar 2022

Dear Dr. Käll,

We are pleased to inform you that your manuscript 'Survival analysis of pathway activity as a prognostic determinant in breast cancer' has been provisionally accepted for publication in PLOS Computational Biology.

Best regards,

Christos A. Ouzounis

Associate Editor

PLOS Computational Biology

Douglas Lauffenburger

Deputy Editor

PLOS Computational Biology

Reviewer's Responses to Questions

**Comments to the Authors:**

Reviewer #1: The authors have made a good attempt to clear up the remaining points. My main worry is still significance / importance to the field, but I leave this to the editor to decide.

1. I agree, there are certainly novel aspects to the paper. However, as I said, the question is their importance/significance. Reflecting on the three aspects the authors point out as novel (e.g. use of sunburst plots), I am not 100% convinced that they represent “High importance to researchers in the field” and “Significant biological and/or methodological insight” as required by the journal criteria.

2. -

3. The authors still feel this point is outside the scope of the manuscript. While I would have liked to have seen more investigation here, I do not see it as a barrier to publication.

4. Thanks for adding the investigation of this point. I think it helps.

5. -

6. OK thanks.

7. OK, thanks.

8. The authors acknowledge that their simulation approach will impact the FDR. It seems strange to me that they do not appear to consider this an issue. I acknowledge that this relates to a different null hypothesis than that used in the main analysis.

9. Ok, thanks.

10. -

11. -

Reviewer #3: No other comments at the current version of manuscript

**Have the authors made all data and (if applicable) computational code underlying the findings in their manuscript fully available?**

Reviewer #1: Yes

Reviewer #3: Yes

PLOS authors have the option to publish the peer review history of their article (what does this mean?). If published, this will include your full peer review and any attached files.

Reviewer #1: No

Reviewer #3: No

---

## [Editor Report · Acceptance letter]

24 Mar 2022

PCOMPBIOL-D-21-01051R2 

Survival analysis of pathway activity as a prognostic determinant in breast cancer

Dear Dr Käll,

I am pleased to inform you that your manuscript has been formally accepted for publication in PLOS Computational Biology. Your manuscript is now with our production department and you will be notified of the publication date in due course.

With kind regards,

Anita Estes
